# An Efficient and Flexible Method for Deconvoluting Bulk RNA-Seq Data with Single-Cell RNA-Seq Data

**DOI:** 10.3390/cells8101161

**Published:** 2019-09-27

**Authors:** Xifang Sun, Shiquan Sun, Sheng Yang

**Affiliations:** 1Department of Mathematics, School of Science, Xi’an Shiyou University, 710065 Xi’an, China; xfangsun@126.com; 2School of Computer Science, Northwestern Polytechnical University, 710072 Xi’an, China; sqsun@nwpu.edu.cn; 3Department of Biostatistics, University of Michigan, Ann Arbor, MI 48109, USA; 4Department of Biostatistics, School of Public Health, Nanjing Medical University, 211166 Nanjing, China

**Keywords:** cell-type compositions, deconvolution, single-cell RNA-seq, nonnegative matrix factorization, gene expression

## Abstract

Estimating cell type compositions for complex diseases is an important step to investigate the cellular heterogeneity for understanding disease etiology and potentially facilitate early disease diagnosis and prevention. Here, we developed a computationally statistical method, referring to Multi-Omics Matrix Factorization (MOMF), to estimate the cell-type compositions of bulk RNA sequencing (RNA-seq) data by leveraging cell type-specific gene expression levels from single-cell RNA sequencing (scRNA-seq) data. MOMF not only directly models the count nature of gene expression data, but also effectively accounts for the uncertainty of cell type-specific mean gene expression levels. We demonstrate the benefits of MOMF through three real data applications, i.e., Glioblastomas (GBM), colorectal cancer (CRC) and type II diabetes (T2D) studies. MOMF is able to accurately estimate disease-related cell type proportions, i.e., oligodendrocyte progenitor cells and macrophage cells, which are strongly associated with the survival of GBM and CRC, respectively.

## 1. Introduction

Accurate measurement of cell types in different tissues can often help our understanding of disease etiology and potentially facilitate the early diagnosis and prevention for complex diseases, especially for cancers [1,2,3]. For example, immune cells, such as CD8+ T cells, often proliferate in special tissues surrounding various types of tumors, mediating the immune response against tumor progression [4]. The traditional bulk samples are measured by the tissue-averaged gene expression levels, resulting in overlooked cellular heterogeneity [5]. The recent advance of single-cell RNA sequencing (scRNA-seq) technologies has allowed us to systemically characterize the heterogeneity of diverse cell types residing in tissues [6,7,8]. However, scRNA-seq studies are expensive and are only limited to a relatively small sample size [9], limiting the ability to investigate cellular heterogeneity across multiple individuals [10]. In contrast, bulk RNA-seq studies measure gene expression profiles for thousands of individuals [11,12,13]. Therefore, developing statistical methods for detecting cell type heterogeneity of existing large-scale bulk RNA-seq data with high-resolution scRNA-seq data plays an important role in understanding the interrelationship between cell type proportions and complex diseases [14].

To date, over sixty computational tools have been developed for deconvolution analysis, and these methods can be casted into two categories: reference-free and reference-based methods [15,16]. Reference-free methods dissect the heterogeneous samples into their constituent cell types with unsupervised schemes, i.e., without any prior reference knowledge [17,18]. For example, posted-modified Negative Matrix Factorization (NMF) directly deconvolutes the gene expression levels of heterogenous samples into the expected expression levels across the cell types and the corresponding cell type proportions using the alternating least square method [19]; Convex Analysis of Mixtures (CAM) identifies subpopulation marker genes from the original mixed gene expressions via convex analysis [20], etc. In contrast, reference-based methods estimate cell type proportions with supervised manner, i.e., with predefined cell type markers [21,22,23]. DeconRNASeq estimates mixed cell proportions with signature matrix though quadratic programming [24]; Cell type of Disease (CoD) quantifies disease-relevant immune cell compositions with a pre-defined list of 61 cell-surface markers based on random forest classification methods [25], etc.

Among these deconvolution methods, both MUlti-Subject SIngle Cell deconvolution (MuSiC) [10] and Cell-type Identification By Estimating Relative Subsets of RNA Transcripts (CIBERSORT) [26] would be able to directly model both bulk RNA-seq data and scRNA-seq data to estimate cell type proportions. However, both methods have several important limitations. First, they model normalized gene expression values and effectively treat observed sequencing data as a continuous outcome. However, the count nature of RNA-seq or scRNA-seq data display high mean-variance dependency [27,28]. Failing to account for the mean-variance dependency of RNA-seq data is known to lead to loss of power in sequencing data analysis [29,30]. Second, both methods treat the cell-type specific gene expression levels which are estimated from scRNA-seq data as reference gene signatures. However, bulk tissues display a heterogeneity of cell compositions across different individuals [10]. Failing to account for the heterogeneity of cell compositions across different individuals may lead to distort the underlying truth biological signals. Third, no tailed methods have been developed to jointly model both bulk RNA-seq data (mixture samples) and scRNA-seq data (cell type-specific expression). Existing approaches often treat calculating cell type-specific gene expression levels and estimating cell type compositions as two separate steps, despite the interconnection between these two different types of analyses [10,26]. Failing to jointly model both types of data may lead to the sub-optimal cell type proportion estimates. In addition, existing methods for signature matrix estimation relies on predefined cell markers or preselected differential expression genes, which may filter out informative genes, resulting in biased estimation of a signature matrix.

Here, we present a new computational tool, Multi-Omics Matrix Factorization (denoted as MOMF), to jointly model bulk RNA-seq data and scRNA-seq data to detect cell type compositions which potentially influence the survival processing effect. MOMF not only directly models raw gene expression counts of both bulk RNA-seq data and scRNA-seq data to avoid the biased cell composition estimations caused by normalization step, but also accounts for the heterogeneity of cell compositions across different individuals to estimate the underlying true cell compositions in bulk tissues. In addition, MOMF is not limited to bulk RNA-seq data and scRNA-seq data which are from the same experiment/study, e.g., bulk RNA-seq data is from The Cancer Genome Atlas (TCGA) database while scRNA-seq data can be from Gene Expression Omnibus (GEO) database. MOMF relies on a nonnegative matrix factorization (NMF) framework [31], using the alternating direction method of multipliers (ADMM) algorithm to infer the parameters in the model [32]. The overview and detailed algorithm of MOMF are shown in Results and Materials and Methods sections, respectively. Finally, we illustrate the benefits of MOMF with three in-depth analyses of real data applications, including the investigation of the relationship between cell compositions and survival statutes in glioblastoma (GBM), the relationship between cell compositions and survival statutes in colorectal cancer (CRC) and the relationship between cell compositions and Hb1Ac level in type II diabetes (T2D). From the results, we found that MOMF is able to accurately estimate two well-known cancer-related cell type proportions, i.e., oligodendrocyte progenitor cells (OPCs) and macrophage cells for the survival of GBM and CRC, respectively.

## 2. Methods and Materials

### 2.1. Model and Algorithm

Here, we jointly model both scRNA-seq data and bulk RNA-seq data to deconvolute mixed bulk RNA-seq data. The schematic view of the MOMF is shown in Figure 1. Specifically, we directly model both scRNA-seq count matrix X and bulk RNA-seq count matrix Y using Poisson distribution, as well as adding a prior distribution to account for the uncertainty of gene expression levels across different individuals in bulk RNA-seq data for parameter estimation. In particular, the gene expression count $Y_{ij}$ for *i*′th individual and *j*′th gene in bulk RNA-seq data, we consider
(1)$$Y_{ij}\;\sim \;Poisson\left(\mu_{ij}^{y}\right),\;i=1,2,\cdots ,\;n_{y};j=1,2,\cdots ,\;p,$$
where $Y_{ij}$ is the number of reads that measure the gene expression levels for *j*′th gene and *i*′th individual; $n_{y}$ is the number of individuals; $\mu_{ij}^{y}$ is an unknoCng mean gene expression level for the *i*′th individual and *j*′th gene; and p is the number of genes; Poisson(·) represents the Poisson distribution.

The gene expression count $X_{kj}$ for *k*′th cell and *j*′th gene in scRNA-seq data, we consider
(2)$$X_{kj}\;\sim \;Poisson\left(\mu_{kj}^{x}\right),k=1,2,\cdots ,\;n_{x};j=1,2,\cdots ,\;p,$$
where $X_{kj}$ is the number of reads that measure the gene expression level for *j*′th gene and *k*′th cell; $n_{x}$ is the number of cells; $\mu_{kj}^{x}$ is an unknown Poisson rate parameter that represents the underlying gene expression level for the *i*′th cell and *j*′th gene; and p is the number of genes; Poisson(·) represents the Poisson distribution.

In above models, we further decompose the unknown parameters $\mu_{ij}^{y}$ and $\mu_{kj}^{x}$ into two low-rank matrices, i.e.,
(3)$$\mu_{ij}^{y}=\sum_{c=1}^{C}\Psi_{ic}W_{cj}+E_{ij}^{y},\;i=1,2,\cdots ,\;n_{y};j=1,2,\cdots ,\;p,$$
where $\Psi_{ic}$ is the cell type-specific proportion for the *i*′th individual and *c*′th cell type; C is the number of cell types.
(4)$$\mu_{kj}^{x}=\sum_{c=1}^{C}\Lambda_{kc}W_{cj}+E_{kj}^{x},k=1,2,\cdots ,\;n_{x};j=1,2,\cdots ,\;p,$$
where $\Lambda_{kc}$ is the low-dimension structure for the *k*′th cell and *c*′th cell type; C is the number of cell type; the parameter $W_{cj}$ is the element in the factor loading matrix that represents the underlying true cell-type specific gene expression level; the factor loading matrix W is shared between bulk RNA-seq and scRNA-seq data, allowing us to jointly model both data types and bypassing the estimation uncertainty inevitably occur in previous deconvolution methods; $E_{ij}^{y}$ and $E_{kj}^{x}$ are the residual terms that account for over-dispersion commonly observed in sequencing studies for bulk RNA-seq data and scRNA-seq data, respectively.

To account for the uncertainty of gene expression levels W in estimation step, we first estimate a reference gene expression panel $h_{cj}$ for each cell type, i.e.,
(5)$$h_{cj}=\frac{\sum_{k\in \Omega_{c}}X_{kj}}{\sum_{k=1}^{n_{x}}X_{kj}},\;c=1,2,\cdots ,C;j=1,2,\cdots ,\;p,$$
where $\Omega_{c}$ is a set of the cells that belong to the cell type c. Then, we modeled the underlying true cell-type specific mean gene expression levels as
(6)$$W_{cj}\sim \;TN\left(h_{cj},\sigma^{2}\right),\;c=1,2,\cdots ,C;j=1,2,\cdots ,\;p,$$
where TN(·,·) denotes the truncated normal distribution to guarantee that the cell type proportions are the non-negative values; the parameter $\sigma^{2}$ is an overall fixed parameter which is estimated from real data to measure the uncertainty. In above model, we are interested in estimating the parameter $\Psi_{ic}$ from bulk RNA-seq data for downstream analyses. The task requires the development of computational algorithms to infer the parameters. To reduce the computational burden of estimation, we used the Alternating Direction Method of Multipliers (ADMM) algorithm which has been widely applied for nonnegative matrix factorization problems [30] to infer the parameters.

To utilize the ADMM algorithm, we first construct the objective function
(7)$$\begin{array}{c} ℒ=D\left(Y|\mu^{y}\right)+Tr\left(U^{y}{\left(\mu^{y}-\Psi W\right)}^{T}\right)+\frac{\rho }{2}\|\mu^{y}-\Psi W{\|}_{F}^{2}+Tr\left(U^{\Psi }{\left(\Psi -\Psi_{+}\right)}^{T}\right)+\\ \frac{\rho }{2}\|\Psi -\Psi_{+}{\|}_{F}^{2}+D\left(X|\mu^{x}\right)+Tr\left(U^{x}{\left(\mu^{x}-\Lambda W\right)}^{T}\right)+\frac{\rho }{2}\|\mu^{x}-\Lambda W{\|}_{F}^{2}+\\ Tr\left(U^{\Lambda }{\left(\Lambda -\Lambda_{+}\right)}^{T}\right)+\frac{\rho }{2}\|\Lambda -\Lambda_{+}{\|}_{F}^{2}+Tr\left(U^{W}{\left(W-H\right)}^{T}\right)+\frac{\rho }{2}\|W-H{\|}_{F}^{2}, \end{array}$$
where $D\left(y|x\right)=ylog\left(\frac{y}{x}\right)-y+x$ is the Kullback-Leibler (KL) divergence; $U^{y},$
$U^{x},$
$U^{\Psi },$
$U^{\Lambda }$ and $U^{W}$ are element-wise coefficients; $\Psi_{+}$ and $\Lambda_{+}$ are the non-negative matrix for Ψ and Λ, respectively; ρ is the penalty parameter; H is reference gene expression panel; W is underlying true gene expression panel; Tr(·) denotes the trace of a matrix. The updating equations for the parameters are as follows:

Taking the derivative of ℒ with respect to $\mu_{ij}^{y}$ and $\mu_{kj}^{x},$ we have
(8)$$\{\begin{array}{c} \mu_{ij}^{y}=\frac{\rho \Psi_{ic}W_{cj}-U_{ij}^{y}-1+\sqrt{{\left(\rho \Psi_{ic}W_{cj}-U_{ij}^{y}-1\right)}^{2}+4\rho Y_{ij}}}{2\rho }\\ \mu_{kj}^{x}=\frac{\rho \Lambda_{kc}W_{cj}-U_{kj}^{x}-1+\sqrt{{\left(\rho \Lambda_{kc}W_{cj}-U_{kj}^{x}-1\right)}^{2}+4\rho X_{kj}}}{2\rho } \end{array}$$

Taking the derivative of ℒ with respect to $\Psi_{ic}$ and $\Lambda_{kc},$ we have
(9)$$\{\begin{array}{c} \Psi ={\left(WW^{T}+I\right)}^{-1}\left(YW^{T}+\Psi_{+}+\frac{1}{\rho }\left(U^{y}W^{T}-U^{\Psi }\right)\right)\\ \Lambda ={\left(WW^{T}+I\right)}^{-1}\left(XW^{T}+\Lambda_{+}+\frac{1}{\rho }\left(U^{x}W^{T}-U^{\Lambda }\right)\right) \end{array}$$

Taking the derivative of ℒ with respect to W, we have
(10)$$W={\left(\Psi^{T}\Psi +\Lambda^{T}\Lambda +I\right)}^{-1}\left(\Psi^{T}Y+\Lambda^{T}X+H+\frac{1}{\rho }\left(\Psi^{T}U^{y}+\Lambda^{T}U^{x}-U^{W}\right)\right)$$

Updating $\Psi_{+},$ and $\Lambda_{+}$ with
(11)$$\Psi_{+}=max\left(\Psi +\frac{1}{\rho }U^{y},0\right),\;\Lambda_{+}=max\left(\Lambda +\frac{1}{\rho }U^{x},0\right)$$

Updating the coefficients $U^{y},$
$U^{x},$ and $U^{W}$ with
(12)$$U^{y}\leftarrow U^{y}+\rho \left(\mu^{y}-\Psi W\right),\;U^{x}\leftarrow U^{x}+\rho \left(\mu^{x}-\Lambda W\right),\;U^{W}\leftarrow U^{W}+\rho \left(W-H\right)$$

### 2.2. Simulation Designs

We performed benchmark experiments to examine the performance of MOMF and compared it with existing approaches, MuSiC and CIBERSORT. The cell type proportion matrix Ψ and the low-dimensional embedding matrix Λ were estimated from CRC data, including 590 individuals of bulk RNA-seq data and 359 cells of scRNA-seq data (details of CRC data in Methods and Materials). Following the model assumption, we first computed the expected gene expression levels of bulk RNA-seq data E(Y)=ΨW and the expected gene expression levels of scRNA-seq data E(X)=ΛW, where W was randomly generated from gamma distribution with shape parameter 2 and inverse scale parameter 2 (i.e., R function *rgamma*). Then, we randomly generated Y and X from Poisson distribution (i.e., R function *rpois*). We simulated 10,000 genes and varied the number of cell types C to be either 2 (Epithelial and Macrophage), 3 (B cell, T cell and macrophage) and 5 (B cell, T cell, Epithelial, Fibroblast, Macrophage) to examine the performance of different deconvolution methods. Finally, we utilized Pearson correlation and mean of difference (MSE) between the estimated proportion $\hat{p}$ to the ground truth p to measure the performance of different methods.

### 2.3. Bulk RNA-Seq and scRNA-Seq Data for GBM

Bulk RNA-seq data of GBM were downloaded from TCGA, which were measured on 56,716 transcripts and 153 individuals. We used the Level-3 Illumina Hiseq data which have performed quality control by TCGA workgroup. All data portals were accessed on March 2016. We filtered out the samples that do not include the survival information or survival time that equals to zero. scRNA-seq data (GSE67835) from brain tissue, consist of 466 cells, including astrocytes (62 cells), endothelial (20 cells), fetal quiescent (110 cells), fetal replicating (25 cells), hybrid (46 cells), microglia (16 cells), neurons (131 cells), oligodendrocytes (38 cells) and OPC (18 cells). We selected the transcripts that are larger than 5 and at least ten cells are expressed for each gene. Finally, we analyzed 144 individuals and 285 cells with common shared 11,120 genes for both bulk RNA-seq data and scRNA-seq data, respectively.

### 2.4. Bulk RNA-Seq and scRNA-Seq Data for CRC

Bulk RNA-seq data of CRC were downloaded from TCGA, measured on 56,716 transcripts and 616 individuals, including 453 Colon Adenocarcinoma (COAD) patients and 163 Rectum Adenocarcinoma (READ) patients. We filtered out the samples that do not include the survival information or survival time that equals to zero. Clinical information of both COAD and READ groups are described in Table 1. We used the Level-3 Illumina Hiseq data which were performed with quality control that was done by the TCGA workgroup. All the data portals were accessed on March 2016. Continuous variables were summarized as mean ± standard deviation (SD), and categorized variables were described by frequency (*n*) and proportion (%). Endpoint is regarded as the death in the survival analysis. We used the log-rank test to compare the survival time of both COAD and READ groups.

scRNA-seq data (GSE81861) from CRC consist of 364 cells, including epithelial (272 cells), fibroblasts (17 cells), endothelial (4 cells), B (17 cells), T (34 cells), mast (1 cell) and macrophage (19 cells) [33]. Finally, we analyzed 590 individuals and 359 cells with common shared 33,888 transcripts for both bulk RNA-seq data and scRNA-seq data, respectively.

### 2.5. Bulk RNA-Seq and scRNA-Seq Data for T2D

We directly downloaded both processed T2D bulk RNA-seq and scRNA-seq data from MuSiC study [10]. For bulk RNA-seq data, we filtered out bulk individuals that do not have Hb1Ac level information. Based on clinical standard, Hb1Ac levels less than 6.0% is classified as normal sample while larger than 6.5% is classified as diabetic sample. For scRNA-seq data, we used five cell types, including the beta cell (171 cells), the delta cell (59 cells), the gamma cell (75 cells), and the ductal cell (135 cells), for downstream deconvolution analysis. Finally, we analyzed 77 individuals and 883 cells with common shared 14,934 transcripts for both bulk RNA-seq data and scRNA-seq data, respectively.

### 2.6. Software for Analyses

All calculations for the pooling of the effect estimates were performed using R program (version 3.6.1). TCGA data were downloaded by *TCGAbiolinks* package (version 2.13.3). The *k*-means clustering algorithm was performed on inferred cell type proportions using R function *kmeans* (iter.max = 10,000). Log-rank tests were performed by *survival* package (version 2.43.1). KM method was performed by *survminer* package (version 0.4.4) to illustrate the survival curves of different clusters. We used *biomaRt* package (version 2.40.1) to transfer Ensembl to gene symbol. MuSiC was performed by *MuSiC* R package (version 0.1.0) with default parameter settings. CIBERSORT was performed by the webserver tool (https://cibersort.stanford.edu/). MOMF requires the raw count gene expression matrices from both bulk RNA-seq and scRNA-seq studies, the penalty parameter ρ of ADMM algorithm was 2 and the number of iterations was 5000. Our method was implemented by the Rcpp as an R package. The source code of all experiments is freely available on GitHub https://github.com/sqsun/MOMF.

## 3. Results

### 3.1. Method Overview

We present a new computational method, MOMF, to estimate the cell type proportions across multiple individuals. An overview of MOMF is provided in Materials and Methods, and the details of the ADMM algorithm are provided in Appendix A. We also illustrate the schematic of the MOMF in Figure 1. Briefly, MOMF jointly models both bulk RNA-seq count matrix Y and scRNA-seq count matrix X to infer the cell compositions Ψ of bulk individuals and low-rank matrix Λ of scRNA-seq data via matrix factorization, i.e., $Y=\Psi W+E^{y}$ and $X=\Lambda W+E^{x},$ where $E^{y}$ and $E^{x}$ represent the residual errors for bulk RNA-seq data and scRNA-seq data, respectively. The common shared gene specific expression matrix W between both bulk RNA-seq data and scRNA-seq data will be inferred from a reference signature expression level to accounting for the heterogeneity of cell compositions across different individuals. Our model starts with scRNA-seq data measured by a few hundreds of thousands cells and assumes that the cell type labels for the cells are known. MOMF deconvolutes the mixture bulk RNA-seq individuals with cell type-specific expression levels to obtain the proportions of the cell types in each individual.

Here, MOMF is able to overcome three drawbacks that MuSiC and CIBERSORT have: (1) it directly models mean-variance dependence of raw gene expression counts, avoiding to introduce systematic errors that may lead to spurious biological signals in downstream analysis; (2) it indirectly models the underlying true signature gene expression levels that follows mean cell type-specific expression levels to account for the heterogeneity of cell compositions across different individuals (i.e., Equation (6) in Methods and Materials), and variance σ^2^ in Equation (6) is to account for the degree of uncertainty; (3) it jointly models the bulk RNA-seq data and scRNA-seq data to avoid the sub-optimal cell type proportion estimates (Methods and Materials). To demonstrate the stability of MOMF, we performed MOMF on the same data. We found that MOMF displays high correlation with two independent runs (i.e., $R^{2}=0.99)$ (Appendix A). Overall, MOMF is a more flexible model to estimate the cell type proportions of bulk RNA-seq data.

### 3.2. Normalization Distorts Raw Expression Counts

As we mentioned in the benefits of MOMF, proper normalization is a critical step that affects the estimation of the cell type proportions in deconvolution analysis. In bulk RNA-seq analysis, normalization has been extensively investigated [28,30,34]. In scRNA-seq data analysis, normalization is still an intractable problem [35,36,37]. The most popular normalization of scRNA-seq data is the log transformation of count per million (CPM) [38], i.e., $\log_{2}\left(y_{ij}/N_{j}+c\right)$ where $y_{ij}$ is the gene expression level for *i*′th cell and *j*′th gene, $N_{i}$ is the read sequencing depth, and c is a pseudocount. The logarithm transformation of CPM (logCPM) may distorts the raw gene expression counts and introduces the zero-inflation artifacts due to the large number of zero counts [39]. For example, all the zeros remain log_2_ (1 + 0) = 0, but the ones turn into value log_2_ (1 + 1/3000 × 10^6^) = log_2_ (334) ≈ 8.4, and the counts that 10 will have value log_2_ (3340) ≈ 11.7. The large, artificial gap between zero and nonzero values makes the log-normalized data appear zero-inflated. To illustrate this phenomenon, we examined the distribution of an example gene (ENSG00000180725) in CRC scRNA-seq data before and after the logarithm transformation with varying normalizations (Appendix A).

### 3.3. Simulations

We first evaluated the performance of different deconvolution methods on simulation studies with ground truth. To do so, we applied three methods, MOMF, MuSiC, and CIBERSORT to simulated datasets (Methods and Materials) and evaluated the performance of different methods based on the Pearson correlation and difference between estimated cell type proportion and ground truth. In the analysis, we varied the number of cell types to be either 2, 3 or 5 to examine their influence on the accuracy of cell compositions. The evaluation results are summarized in Figure 2, Appendix A.

From the results, we found that MOMF achieves the best performance across all parameter settings. For example, with the simulated data based on three cell types (B cells, T cells and Macrophage cells), the Pearson correlation R generated by MOMF is 0.992 and MSE is 0.040, while MuSiC (R is −0.753 and MSE is 0.611) and CIBERSORT (R is −0.213 and MSE is 0.334) do not fare well (Figure 2). With the small number of cell types (i.e., 2), all three methods are able to generate high Pearson correlations (Appendix A). However, the scatter plots show the vertical patterns due to many zeros or ones and proportions were estimated by MuSiC and CIBERSORT. The Pearson correlation will decrease when increasing the number of cell types (i.e., 5). For example, the Pearson correlation R generated by MOMF is 0.618 and MSE is 0.135, while MuSiC (R is −0.368 and MSE is 0.490) and CIBERSORT (R is −0.692 and MSE is 0.395) do not fare well (Appendix A).

### 3.4. Human Glioblastoma (GBM) Data

GBM is the most common primary brain tumor in adults [40,41,42]. GBM is characterized by the presence of hyperplastic blood vessels and the presence of small areas of necrotizing tissue that are surrounded by anaplastic cells [43]. Therefore, our primarily goal here is to characterize how cell type proportions influence the survival time of GBM. We first applied MOMF on bulk RNA-seq data from GBM, which consist of 153 individuals and 60,486 transcripts and scRNA-seq data from healthy human brains, which consist of nine cell types and 18,752 transcripts (details in Materials and Methods). These cell types include astrocytes (62 cells), endothelial (20 cells), fetal quiescent (110 cells), fetal replicating (25 cells), hybrid (46 cells), microglia (16 cells), neurons (131 cells), oligodendrocytes (38 cells) and OPC (18 cells) [44,45]. Following the preprocessing (Materials and Methods), for bulk RNA-seq data, we finally performed the analyses on 144 individuals along with median survival time (MST) 333 days; for scRNA-seq data, we finally performed the analyses on 285 cells from six different cell types, including astrocytes (62 cells), endothelial (20 cells), microglia (16 cells), neurons (131 cells), oligodendrocytes (38 cells) and OPC (18 cells). Finally, 11,120 transcripts commonly shared between both bulk RNA-seq data and scRNA-seq data were performed in the experiments.

We applied three deconvolution methods, MOMF, MuSiC, and CIBERSORT on both datasets to estimate the cell compositions across all GBM individuals. Average cell type proportions of astrocytes, endothelial, microglia, neurons, oligodendrocytes and OPC estimated from MOMF are roughly 3.6%, 25.2%, 4.9%, 4.7%, 3.2% and 57.9%, respectively; from MuSiC are 20.3%, 32.3%, 14.6%, 28.3%, 4.5% and 0.0%, respectively; and from CIBERSORT are 31.5%, 27.4%, 12.8%, 6.6%, 8.8% and 12.4%, respectively (Figure 3A). From the results, we found that MOMF provided higher OPCs cell type proportion (57.9%) than MuSiC (0.0%) and CIBERSORT (12.4%). OPCs are highly associated with cellular differentiation for oncogenic transformation in RCAS/tv-a model [41] and can be broadly arrayed in an adult brain where they constitute the largest pool of dividing cells [46]. The second-high cell type proportion is from endothelial cells. The cell type proportions from three methods MOMF, MuSiC, and CIBERSROT are 25.2%, 32.3% and 27.4%, respectively. The connection between neural stem cells and the endothelial compartment plays an important role in GBM. A significant proportion of the vascular endothelium has a neoplastic origin with increasing endothelial cells [47,48]. The high cell type proportions of OPCs might not be necessary to show the high performance of MOMF due to the ground truth of cell type proportion is being unknown. To validate the cell type proportions estimated from three different methods, we performed the association between the cell type proportion and the survival time of its individuals as another criterion to measure the performance of different deconvolution methods. MOMF produced statistically significant Kaplan-Meier (KM) plot with four potential subtypes (*N_CL1_* = 57, *N_CL2_* = 33, *N_CL3_* = 44, *N_CL4_* = 10) of GMB cancer (*p*-value = 0.007, log-rank test) (Figure 3B). With short median survival time, the cluster CL4 identified by MOMF has poor-prognosis for subtype of GBM cancer. While the *p*-value provided by MuSiC and CIBERSORT are 0.065 (*N_CL1_* = 60, *N_CL2_* = 34, *N_CL3_* = 19, *N_CL4_* = 31) and 0.170 (*N_CL1_* = 52, *N_CL2_* = 39, *N_CL3_* = 26, *N_CL4_* = 27), respectively (Figure 3B), failing to identify the potential subtype of GMB cancer.

### 3.5. Human Colorectal Cancer (CRC) Data

We next applied MOMF on CRC RNA-seq data, which consist of gene expression measurements from 56,716 transcripts and 616 individuals; scRNA-seq data which were measured on 364 cells from seven subpopulations (details in Materials and Methods). These seven subpopulations include T cells (34 cells), B cells (17 cells), epithelial cells (272 cells), fibroblast cells (17 cells), macrophage cells (19 cells), endothelial (4 cells), mast (1 cell) [33,49]. Following the preprocessing (Materials and Methods), for bulk RNA-seq data, we analyzed 590 individuals along with a median survival time (MST) of 2,532 days; for scRNA-seq data, we finally analyzed 359 cells of five cell types, T cells (34 cells), B cells (17 cells), epithelial cells (272 cells), fibroblast cells (17 cells) and macrophage cells (19 cells). Finally, we examined 33,888 common shared transcripts on both bulk RNA-seq data and scRNA-seq datasets.

To systematically benchmark the performance of MOMF, MuSiC and CIBERSORT, we applied them to CRC studies to estimate the cell-type proportion of bulk RNA-seq data across all individuals. We first applied MOMF, MuSiC and CIBERSORT to estimate the cell type proportions of CRC bulk individuals (Figure 4A). From the results, we found that MOMF provides a higher macrophage cell proportion (14.9%), rather than MuSiC (2.0%) and CIBERSORT (5.1%). Tumor-associated macrophages (TAMs) are important components of the tumor microenvironment [50]. Macrophage cells may contribute to tumor growth and progression by promoting tumor cell proliferation and invasion, fostering tumor angiogenesis and suppressing antitumor immune cells [51]. The second high contribution of cell type proportion is the epithelial cell subpopulation (Figure 4A). The cell type proportions from MOMF, MuSiC and CIBERSORT are 49.7%, 90.0% and 73.6%, respectively. Small and large intestinal epithelium culture in vitro shows prolonged intestinal epithelial expansion with proliferation and multilineage differentiation [52]. The high cell type proportions of macrophage cells might be not necessary to show the high performance of MOMF due to the ground truth of cell type proportion is unknown. To validate the cell type proportions estimated from three different methods, we further performed the association analysis between the cell type proportion and its survival data, and the KM plot shows median survival time, number at risk and number of censoring of clustering results from the three methods. MOMF produced statistically significant KM plot with four potential subtypes (*N_CL1_* = 3, *N_CL2_* = 90, *N_CL3_* = 277, *N_CL4_* = 220) of CRC (*p*-value = 0.0013, log-rank test) (Figure 4B). With short median survival time, the cluster CL1 identified by MOMF has poor-prognosis for the subtype of CRC. While the *p*-value generated by MuSiC and CIBERSORT are 0.31 (*N_CL1_* = 23, *N_CL2_* = 241, *N_CL3_* = 228, *N_CL4_* = 98) and 0.098 (*N_CL1_* = 246, *N_CL2_* = 113, *N_CL3_* = 27, *N_CL4_* = 204), respectively (Figure 4B).

### 3.6. Human Type II Diabetes (T2D) Data

We finally applied MOMF on pancreatic islet studies: the bulk RNA-seq data consist of gene expression measurements of 32,581 transcripts and 89 individuals; the scRNA-seq data consist of 25,453 genes and 1097 cells from 14 subpopulations (details in Materials and Methods). The 14 subpopulations include alpha (443 cells), beta (171 cells), delta (59 cells), gamma (75 cells), ductal (135 cells), acinar (112 cells), co-expression (26 cells), endothelial (13 cells), epsilon (5 cells), mast (4 cells), MHC class II (1 cell), PSC (23 cells), unclassified (1 cell) and unclassified endocrine (29 cells) [33,49]. Following the preprocessing (Materials and Methods), for bulk RNA-seq data, we totally analyzed 77 individuals along with hemoglobin A1c (HbA1c) level; for scRNA-seq data, we analyzed 883 cells from five cell types, including 443 alpha cells, 171 beta cells, 59 delta cells, 75 gamma cells and 135 ductal cells. We examined 14,934 common shared genes on both bulk RNA-seq data and scRNA-seq data.

We first applied MOMF, MuSiC and CIBERSORT on pancreatic islets studies to recover the cell type proportion of bulk RNA-seq data across all individuals. Because bulk RNA-seq data contain T2D and control individuals, we examined the cell type proportions separately. From the results, we found that MOMF generated high ductal cell type proportions across T2D bulk individuals, and the percentage of ductal cell in T2D individuals is higher than that in normal individuals/controls (Figure 5A). Specifically, the recovered cell type proportions of MOMF, MuSiC and CIBERSORT are 97.4%, 53.1% and 35.8% in T2D individuals and 96.3%, 39.9% and 22.9% in controls, respectively. From the previous studies [53], our model validated the findings that the increased pancreatic ductal replication is strongly associated with T2D. The variance of cell type proportions estimated from MuSiC and CIBERSORT are extremely large. It is presumably due to both methods that are not able to account for the heterogeneity of cell compositions across different individuals.

To further validate the effectiveness of MOMF, MuSiC and CIBERSORT, we performed the association between cell type proportions and HbA1c levels. With cell type proportion results, we performed simple linear regression (*lm* function in R) with HbA1c levels, controlling for gender, age and body mass index (BMI) as the covariates. From the results, we found that the beta cell proportion shows the negative correlation with HbA1c levels while ductal cell proportion displays the positive correlation with HbA1c levels (Figure 5B). Both MOMF and MuSiC show the strong association with beta cell proportions, i.e., *p*-value = 0.004 and *p*-value = 0.006, respectively (Figure 5B), while CIBERSORT is weak, i.e., *p*-value = 0.683 (Figure 5B). A combination of increasing insulin resistance and reduced mass or dysfunction of the beta cells is the potential factor of T2D [54].

## 4. Discussion

In this paper, we presented a new deconvolution method, MOMF, which directly models raw sequencing count data and accounts for the heterogeneity of cell compositions across different individuals. We have illustrated the benefits via performing deconvolution analysis through MOMF on both sequencing data (bulk RNA-seq and scRNA-seq). We have shown that MOMF is the only method currently available that can jointly model bulk RNA-seq and scRNA-seq data, providing the accurate measurement of cell type proportions which are highly associated with its corresponding survival times. With three in-depth analyses of real data applications, MOMF displays more reasonable and convincing results than existing two deconvolution methods, MuSiC and CIBERSORT. In contrast, MuSiC does not perform well on rare cell type proportion estimates, i.e., the inferred cell type proportions from bulk RNA-seq are way off (very close to zero). Overall, MOMF is a useful and efficient tool, which is implemented as an R package for analyzing cell type compositions in tissue-specific gene expressions.

We primarily focused on using both RNA-seq data that are all modelling raw gene expression counts. One of the potential applications of MOMF is to deconvolute the mixed gene expression data which are from microarray experiments (i.e., continuous data). Therefore, exploring MOMF for continuous-counts or continuous-continuous scenarios is probably going to be part of our further work. In this case, MOMF will be useful to analyze the gene expression datasets. It can be easily extended to other multi-omics data analyses which can be either continuous data or count data [55,56]. This will probably be a very promising research direction in our further work.

MOMF is not without limitations. Perhaps the biggest limitation of MOMF is that the cell types are required to be labeled before applying MOMF. For scRNA-seq data, where there is no cell type label information, one solution is to first utilize the existing clustering methods, such as scNBMF or Seurat [57,58], to specify the cell type label for each cell, and then run MOMF with labeled scRNA-seq data.

## Figures and Tables

**Figure 1 cells-08-01161-f001:**
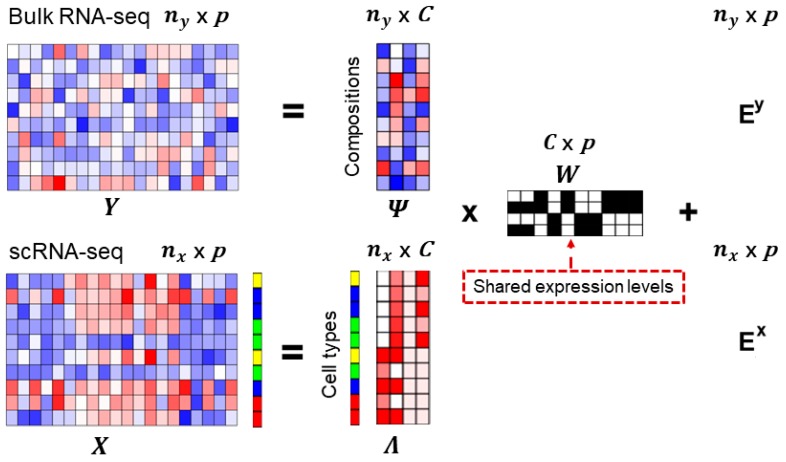
Overview of Multi-Omics Matrix Factorization (MOMF) framework. MOMF integrates bulk RNA-seq data and scRNA-seq data, to deconvolute the two expression matrices by the shared information and estimate the cell-type proportions for each individual. Specifically, MOMF jointly models both bulk RNA-seq count matrix Y and scRNA-seq count matrix X to infer the cell compositions Ψ of bulk RNA-seq data and low-rank matrix Λ of scRNA-seq data via matrix factorization, i.e., $Y=\Psi W+E^{y}$ and $X=\Lambda W+E^{x},$ where W is the common shared gene expression levels and $E^{y}$ and $E^{x}$ represent the residual errors for bulk RNA-seq data and scRNA-seq data, respectively. The heatmaps are used to illustrate the gene expression level (Y and X); cell specific expression levels (bulk RNA-seq: Ψ; scRNA-seq: Λ); and gene specific expression levels (W). The color bar along with the heatmaps of scRNA-seq data represents the cell types. $n_{y}$ is the number of individuals; $n_{x}$ is the number of cells; p is the number of common shared genes; C is the number of cell types.

**Figure 2 cells-08-01161-f002:**
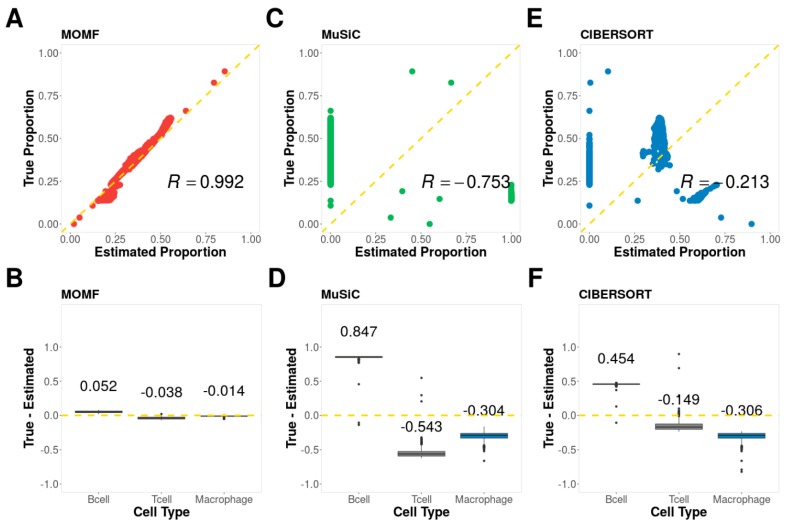
Simulation results. The simulated data based three cell types, B cells, T cells, and Macrophage cells. (**A**) The scatter plot of ground truth and cell type proportion estimated by MOMF; (**B**) The boxplot to show the difference between ground truth and cell type proportion estimated by MOMF (**C**) The scatter plot of ground truth and cell type proportion estimated by MuSiC; (**D**) The boxplot to show the difference between ground truth and cell type proportion estimated by MuSiC (**E**) The scatter plot of ground truth and cell type proportion estimated by CIBERSORT; (**F**) The boxplot to show the difference between ground truth and cell type proportion estimated by CIBERSORT. R: Pearson correlation.

**Figure 3 cells-08-01161-f003:**
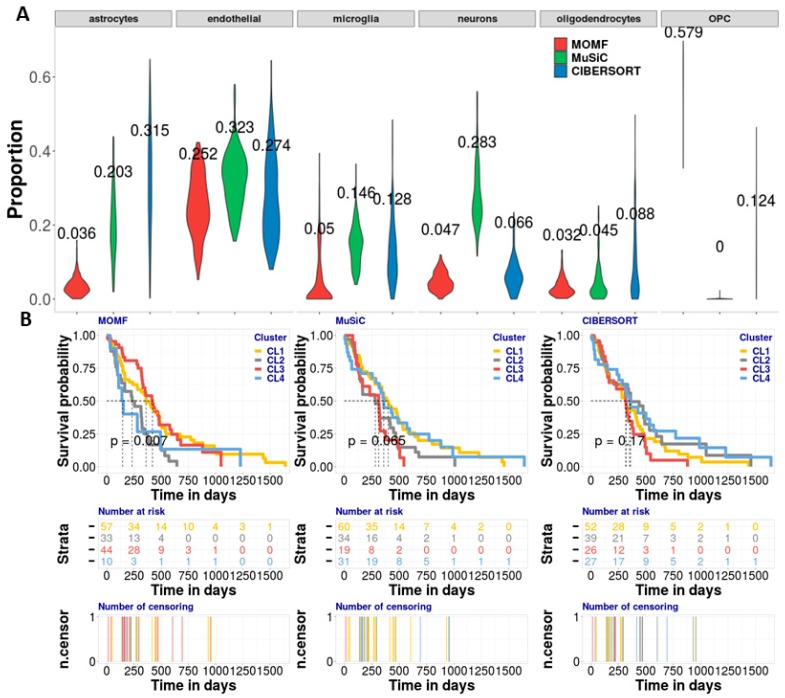
Analyzing GBM bulk RNA-seq data with brain scRNA-seq data. (**A**) The violin plot is to show the effect proportion of each cell type from three different deconvolution methods. We found that the OPCs and endothelial cells are enriched by MOMF, which means that the two cell types potentially contribute to the survival of GBM. (**B**) The KM plots are used to show the survival analysis for four clusters from the TCGA bulk RNAseq data. We use log-rank test to compare the distributions of four clusters. MOMF grouped the GBM samples into four subtypes (*p*-value = 0.007). The cluster CL4 is the poor-prognosis. Number at risk in the table shows the number of survival individuals at each 250 days. Number of censoring in the table shows censoring time of each individual. The numbers labeled in different colors in the two tables indicate the different subtypes.

**Figure 4 cells-08-01161-f004:**
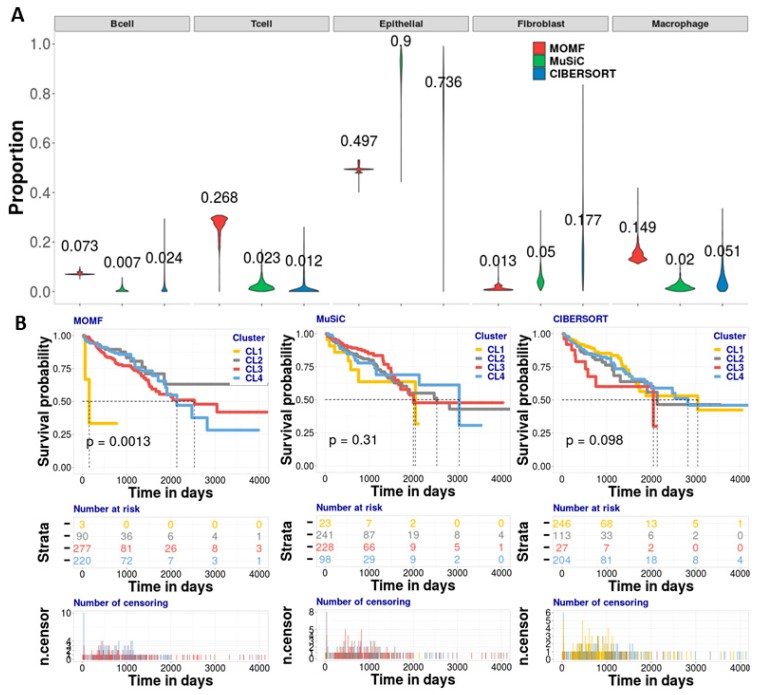
Analyzing CRC bulk RNA-seq data with colorectal cancer scRNA-seq data. (**A**) The violin plot is to show the effect proportion of each cell type from three different methods. We found that the epithelial, T and macrophage cells are enriched by MOMF, which means that the two cell types potentially contribute to the survival of CRC. (**B**) The KM plots are used to show the survival analysis for four clusters from the TCGA bulk data. We used log-rank test to compare the distributions of four clusters. MOMF grouped the CRC samples into four subtypes (*p*-value = 0.0013). The cluster CL1 shows the poor-prognosis. Number at risk in the table shows the number of survival individuals at each 1,000 days. Number of censoring in the table shows censoring time of each individual. The numbers labeled in different colors in the two tables indicate the different subtypes.

**Figure 5 cells-08-01161-f005:**
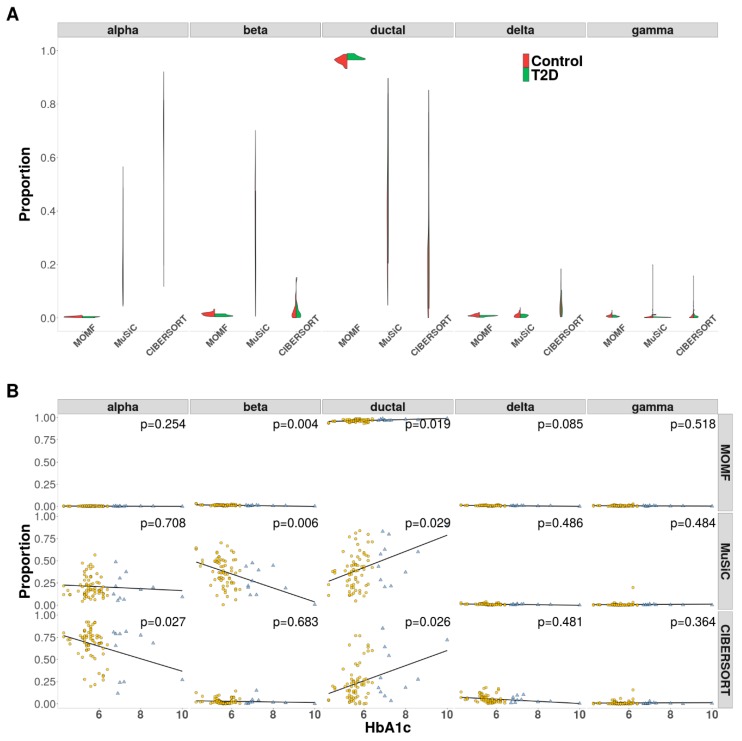
AnalyzingT2D bulk RNA-seq data with pancreatic scRNA-seq data. (**A**) The violin plot is to show the effect proportion of each cell type from three different methods. beta and ductal cells are enriched from MOMF, which means that the two cell types potentially contribute to the survival of CRC. (**B**) The scatter plots are used to show the results of the associations between Hb1Ac level and cell proportion as adjust the covariates. The estimated beta cell proportions by both MOMF and MuSiC are strongly associated with Hb1Ac (*p*-value = 0.004 and 0.006).

**Table 1 cells-08-01161-t001:** Demographic distribution of discovery and validation study populations.

Variables	COAD (*N* = 435)	READ (*N* = 155)	*p*
Age (years), mean ± SD	67.30 ± 12.97	65.33 ± 11.49	0.089
Gender, n (%)			0.680
Female	202 (46.43)	69 (44.52)	
Male	233 (53.56)	86 (55.48)	
Tumor stage (%)			0.166
0-I	240 (55.17)	76 (49.68)	
II-IV	184 (42.30)	71 (45.81)	
Unknown	11 (2.53)	8 (5.16)	
Race (%)			7.25 × 10^−4^
White	207 (45.59)	77 (47.59)	
Non-white	70 (41.08)	6 (6.5)	
Unknown	158 (13.33)	71 (45.81)	
Survival year (month)			
Median	2532	1741	0.3
Dead, n (%)	97 (22.23)	25 (16.13)

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
