# Peer review of "An Efficient and Flexible Method for Deconvoluting Bulk RNA-Seq Data with Single-Cell RNA-Seq Data"

_cells, 2019, doi:10.3390/cells8101161_

Round 1
Reviewer 1 Report
The manuscript ‘An efficient and flexible method for deconvoluting bulk RNAseq data with single-cell RNAseq data’ provides a new method (MOMF) for accurate estimation of cell type proportions. Overall, this manuscript still needs to be revised for English and grammar before it is considered further for publication.
While the authors have completed some edits for the manuscript since the last version, the entire manuscript still needs to be reviewed for edits/English language/structure/grammar. Examples include but are not limited to:
Line 33: What does ‘limiting to investigate’ mean? Do the authors mean ‘limiting the ability to investigate’?
Line 34: Remove ‘the’ before ‘bulk’.
Line 74: What is ‘biases estimation’? Do the authors mean ‘biased estimation’?
Line 77: Remove ‘on’.
Line 83: The sentence does not state what MOMF allows. This is confusing.
Line 104: Should this be ‘deconvolutes’ instead of ‘deconvolves’?
Figure 1 still has elements that are not well described in the figure legend or text of the manuscript. What are the symbols that are in the dashed box to the left of the heat maps for Y and X?
Overall, the manuscript is still has issues with editing. It is not the job of the reviewer or editor to completely edit the manuscript.
Author Response
-------------------
Reviewer: 1#
General Comment:
The manuscript ‘An efficient and flexible method for deconvoluting bulk RNAseq data with single-cell RNAseq data’ provides a new method (MOMF) for accurate estimation of cell type proportions. Overall, this manuscript still needs to be revised for English and grammar before it is considered further for publication.
Thank you very much for the constructive comments and valuable suggestions, which have significantly improved the quality of the paper. We have polished the English and grammar in the updated version. Responses to your specific comments are listed below:
Specific comments:
While the authors have completed some edits for the manuscript since the last version, the entire manuscript still needs to be reviewed for edits/English language/structure/grammar.Line 33: What does ‘limiting to investigate’ mean? Do the authors mean ‘limiting the ability to investigate’?
Line 34: Remove ‘the’ before ‘bulk’.
Line 74: What is ‘biases estimation’? Do the authors mean ‘biased estimation’?
Line 77: Remove ‘on’.
Line 83: The sentence does not state what MOMF allows. This is confusing.
Line 104: Should this be ‘deconvolutes’ instead of ‘deconvolves’?
Figure 1 still has elements that are not well described in the figure legend or text of the manuscript. What are the symbols that are in the dashed box to the left of the heat maps for Y and X?
Response: Thank you for the comments. Following your suggestion, we have carefully revised all sentences throughout the manuscript. Sorry for ambiguous demonstration in Figure 1. We previously used the photos in dashed boxes to show the tissue in bulk RNA-seq and the cell types in scRNA-seq data. For simplicity, we have removed these photos in the updated figures.
Reviewer 2 Report
Please see the attached file.

Author Response
Reviewer: 2#
General Comment:
This paper proposes a statistical method MOMF that considers both bulk RNA sequencing data and single-cell RNA sequencing data to estimate cell-type compositions. The method is tested on three real applications to show its advantages. The writing and organization of the paper are good and easy to understand. The proposed method MOMF is also technically sound. The paper can be accepted if the following concerns can be addressed further.
Thank you very much for the constructive comments and valuable suggestions, which have significantly improved the quality of the paper. All revisions are marked with “Track Changes” in Word. Responses to your specific comments are listed below:
Specific comments:
As described in the second paragraph of page 2, there are three limitations for previous two methods. However, it is not clear why the MOMF can overcome the three limitations. An important characteristic of MOMF is that it can avoid the biases caused by normalization step, as given in the third paragraph of page 2. Some experiments should be done to show this characteristic.
Response: Sorry for the unclear statements. We highlighted the solution of the three limitations that MOMF can overcome in Method overview section (lines 92-100 on page 6). For the normalization, we added one section to explain the issue caused by normalization step in RNA-seq data analysis (Supplementary Figure S2; lines 104-119 on page 6-7).
The experimental results of the three applications are not totally convincing. Take the first application GBM given on page 5 as an example. The authors declared that MOMF is the best because it assigns a larger proportion to cell type OPCs with respect to the other two methods. However, CIBERSORT has the largest proportion on cell type astrocytes, and hence can we say CIBERSORT is the best? The other two applications use the same logic to draw conclusions.
Response: Thanks for the comments. We totally agree with you that the high proportion to a particular cell type is not necessary to show the good performance for the deconvolution method. Because there is no ground truth of cell type proportions, in each data application, we first examined the cell type proportion recovered by different deconvolution methods, and then try to interpret how the cell type proportion related to the particular study (e.g., GBM, CRC, etc.). We feel that this evaluation is likely to be unsatisfactory. Therefore, we further performed the association analysis between the cell type proportions and survival time or phenotypes to convince the readers. We reorganized these ambiguous statements in the updated manuscript (lines 172-180 on page 9 for GBM; lines 209-215 on page 10 for CRC; lines 248-258 on pages 11-12 for T2D). We do hope the reviewer would agree with us that the convergence support from all the above evidence.
To evaluate the performance of different deconvolution methods, we designed three simulation scenarios that contain the ground truth, i.e., the true cell type proportions. We applied three method MOMF, MuSiC, and CIBERSORT on simulated data sets. MOMF displays the highest Pearson correlation and the lowest MSE. The results are summarized in Figure 2, and Supplementary Figures 3 and 4, with details explained in the Results section (lines 120-140 on pages 7-8).
In line 222 of page 7, the number of cell types is 5, not 6
Response: Thanks for your comments. We have revised this error (lines 232 on page 11) in the updated version.
This manuscript is a resubmission of an earlier submission. The following is a list of the peer review reports and author responses from that submission.
Round 1
Reviewer 1 Report
Multi-Omics Matrix Factorization (MOMF) proposed in this manuscript jointly models bulk RNA-seq data and scRNA-seq data to detect cell type compositions. MOMF models the problems as a nonnegative matrix factorization and uses alternating direction method of multipliers (ADMM) algorithm for inferring the parameters. The manuscript is well written but needs a few minor corrections.
The authors have supplied a Github URL for the repository. Please create a release for this paper and generate a DOI for it using Zenodo so that the scripts are archived and citable.
This manuscript needs to be edited by a/another native English speaker as there are numerous minor grammatical and typographical errors that can be easily remedied. The sentence construction can also be improved in some cases.
Fig 2 A: Proportion
Ln 76: which are potentially associated to complex disease.
Ln 247: where there are no cell type
Reviewer 2 Report
The manuscript ‘An efficient and flexible method for deconvoluting bulk RNAseq data with single-cell RNAseq data’ provides a new method (MOMF) for accurate estimation of cell type proportions. Overall, this manuscript does not read well and needs to be revised for English and grammar before it is considered further for publication. Further detail in the materials and methods needs to be added to define data sets. The specific points that resulted in this conclusion are listed below.
Entire manuscript needs to be reviewed for edits/English language/structure/grammar. Examples include but are not limited to:
Lines 28, 30, 42: Spaces need to be removed between end of sentence and period.
Line 40: Space needs to be removed between analysis and comma.
Lines 127 and 128: ‘we finally analyzed on’ needs to be changed to ‘analysis was completed on’.
Line 208 and 209: ‘There is an evidence show that’ needs to be replaced with ‘Evidence shows that’.
Figures along with the corresponding legend need to stand alone. Figure 1 is not well described in the figure legend or text of the manuscript. For example, what do the symbols to the left of each heat map (assuming these are heat maps since this is not specified in legend) represent? This needs to be revised.
Figure 2 and 3: It is difficult to read and interpret the number at risk and number of censoring in part B. The print is too small. This needs to be revised.
It is not initially clear what the three data sets examined were. Additionally, the explanations of these individual data sets are not clear. Line 122-123: are the 18,752 transcripts from the 153 individuals for bulk RNA-seq data? How many reads are included in the 9 subpopulations? Descriptions of all data sets need to be clarified and expanded on.
Lines 132: What are the ‘both data sets’ that the authors are referring to? These data sets are not defined clearly.
Line 193: What are 32,581 variables? Are these transcripts?
Lines 44-49: NMF, CAM and CoD need to be defined.
Overall, the review is not well written and needs to be edited thoroughly before considered for publication. It is not the job of the reviewer to completely edit and rewrite the manuscript. This manuscript needs to be completely overhauled before considered for acceptance.
Reviewer 3 Report
I did not complete the reading of the paper. The reason is that the quality of the presentation of the paper is very low and unacceptable. The authors did not take seriously the submission. Here are just a few examples:
There are no explanations of the symbols in Figure 1. In Figures 2,3 and 4, the name of Y-axis of A should be "Proportion". The quality of these figures are very bad. Line 263 It looks no k in the formulas 2,3 and 4. For other formulas as well, the authors need to be consistent with variable labels as it confuses the readers (especially with k, i, and c) no formula number after formula 4. Typo in Table 1, under ‘Tumor stage (%)’ and ‘Race (%)’, ‘unknow’ should be ‘unknown’ Labels are misleading - in the text and figure captions, for CRC MOMF NCL4 = 3, but in Figure 3B NCL2 = 3 and has poorest survival... There are many others.